# AGING WITH GRACE: LIFELONG MODEL EDITING WITH DISCRETE KEY-VALUE ADAPTORS

## ABSTRACT

Large pre-trained models often err during deployment as input distributions shift, user requirements change, or crucial knowledge gaps are discovered. Recently, model editors have been proposed to modify a model's behavior by adjusting its weights during deployment. However, when editing the same model multiple times, these approaches quickly decay a model's performance on upstream data and forget how to fix previous errors. We propose and study a novel Lifelong Model Editing setting, where streaming errors are identified for a deployed model and we update the model to correct its predictions without influencing unrelated inputs. We introduce General Retrieval Adaptors for Continual Editing, or GRACE, which learns to cache a chosen layer's activations in a codebook as edits stream in, while the original model weights remain frozen. GRACE succeeds to edit pre-trained models thousands of times in a row using *only* streaming errors, while minimally influencing unrelated inputs. Experimentally, we show that GRACE substantially improves over recent model editors while generalizing to unseen inputs.

## 1 INTRODUCTION

Modern machine learning systems perform extremely well on challenging, real-world tasks. Many successes stem from large models trained on massive amounts of data, achieving state-of-the-art performance on challenging tasks in natural language processing (Rae et al., 2021; Brown et al., 2020) and computer vision (Ramesh et al., 2022; Dosovitskiy et al., 2020). However, despite high performance, large models still make critical mistakes during deployment (Sinitsin et al., 2019; Balachandran et al., 2022; Wang et al., 2021). Further, when models are deployed over long periods of time without updates, their error rates increase as data distributions shift, as labels shift, or as ground-truth information about the world simply changes (Lazaridou et al., 2021). For example, a language model trained in 2016 that correctly identifies Barack Obama as president of the United States would be incorrect after 2017, as illustrated in Figure 1. Furthermore, multiple errors often occur sequentially, requiring many fixes to the same model over time. To handle this case, we introduce *Lifelong Model Editing* to continuously correct large models' mistakes over long sequences of edits. By *editing* a model, we avoid costly retraining while maintaining its performance on unrelated inputs (Mitchell et al., 2022a).

One approach to lifelong editing is to directly finetune a model on errors as they arrive. However, finetuning on errors is prone to overfitting, even with customized regularization (Lin et al., 2022). Further, finetuned models quickly forget original training data, devaluing upstream pretraining (Lee et al., 2020). And even worse, finetuned models can easily forget previously-fixed errors, counteracting the objective of editing in the first place (Sinitsin et al., 2019).

Another candidate for lifelong editing is through standard model editing, which outperforms finetuning by updating model behavior with minimal influence on unrelated inputs (Mitchell et al., 2022a). While a promising direction, these methods require large amounts of data in order to make edits. For example, some recent works use training sets filled with pre-collected errors to train hypernetworks (von Oswald et al., 2020) that edit a model's behavior by predicting new weights (Mitchell et al., 2022a) or offload predictions to a new classifier (Mitchell et al., 2022b). However, large pools of representative training edits are rarely available before deployment. Alternatively, regularized finetuning approaches (Meng et al., 2022a; Sinitsin et al., 2019; De Cao et al., 2021) rely on sets of semantically-equivalent inputs to preserve upstream model performance. Additionally, prior model

Continually Editing a Pre-Trained model with **GRACE**

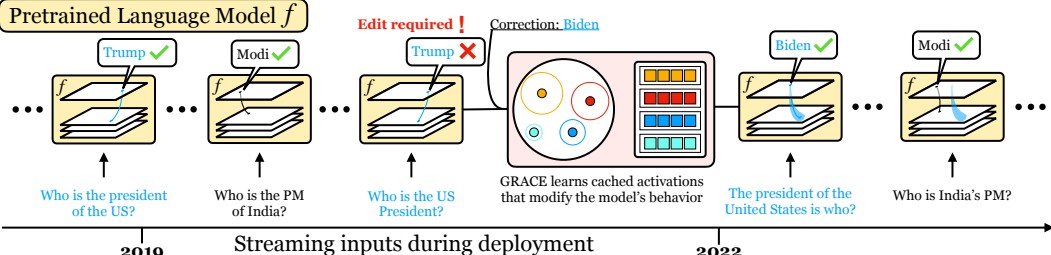

Figure 1: GRACE edits pretrained models during deployment by maintaining layer-specific codebooks that cache learned activations for selected layers. When inputs similar to previous edits arrive, the corresponding activations are passed to the next layer to correct the model's predictions. GRACE continuously edits a deployed model using *only* streaming errors without accessing upstream data.

editors (Mitchell et al., 2022a; Sinitsin et al., 2019; De Cao et al., 2021; Mitchell et al., 2022b) fail to make multiple edits sequentially. Consequently, they fail as lifelong editors, as we demonstrate.

In this paper, we introduce *Lifelong Model Editing*. Assume we have a model $f_0$ that was pretrained on a set of upstream instances $\mathcal{U}$. During deployment, we observe a stream of errors $\{(X_t^e, y_t^e)\}_{t=1}^T$ such that $f_{t-1}(X_t^e) \neq y_t^e \, \forall \, t$. Here, $f_{t-1}$ is the model being used for inference at step $t$, and $y_t^e$ is the *correct* label for input $X_t^e$. At each step $t$, we receive an input–label pair for an edit $(X_t^e, y_t^e)$, and our aim is to produce an edited model $f_t$ that meets three criteria:

1. $f_t(X_t^e) = y_t^e$; the error at step $t$ is corrected.

2. $f_t(X_i^e) = y_i^e$ for $i \leq t$; the model remembers the corrections for previous errors.

3. $f_t(X) = y \, \forall \, (X, y) \in \mathcal{U}$; the model's upstream test performance is maintained.

To address this challenging setting, we propose General Retrieval Adaptors for Continual Editing, or GRACE. GRACE edits individual layers of a frozen, pretrained model $f_0$, treating it as an encoder. GRACE introduces a codebook memory to a chosen layer, using encodings from the previous layer as queries to find the nearest key, which is associated with a value. Each value can replace the model's predicted activation, which can be decoded by future layers, ultimately leading to a prediction. Additionally, GRACE learns one $\epsilon$-ball per key. During inference, if an encoding does not land within any key's $\epsilon$-ball, then the frozen pre-trained model is used directly, avoiding interference with any edits. By adding and updating key–value pairs and their $\epsilon$ values as edits stream in, GRACE fixes model mistakes without decaying performance on upstream training data. For example, in Figure 1, where GRACE corrects $f$'s predictions about the United States President without impacting knowledge of India's Prime Minister. During training GRACE adapts to changing distributions of edits by shrinking and expanding each $\epsilon$-ball, leading to coarse- or fine-grained influence on the edited layer's representation space. Further, we can initialize $\epsilon$ to manage the trade-off between remembering upstream data and successfully fixing long sequences of errors.

Our contributions in this work are:

1. We cast model editing in a new and realistic lifelong streaming setting. To our knowledge, this setting is unstudied, yet is crucial to successfully deploying large language models.

2. We present GRACE, a novel key-value model editor which learns to cache and retrieve activations for selected layers *using only errors observed during deployment*. Further, GRACE directly applies to any existing transformer-based model, with straight-forward alterations for other architectures.

3. Our experiments show that GRACE is a state-of-the-art model editor, outperforming alternatives like MEND (Mitchell et al., 2022a) on domain-free question answering and a classification task with shifting label distributions. We also find that GRACE successfully generalizes edits to previously-unseen inputs.

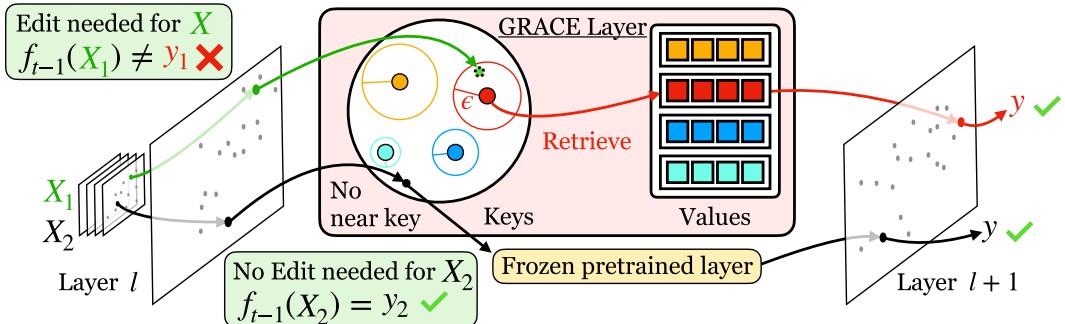

Figure 2: GRACE learns and maintains a codebook for selected layers in a pretrained model. Each codebook contains keys and their corresponding $\epsilon$ and values. When an input like $X_1$ is similar to any previous edits, the GRACE codebook produces an activation to correct the behavior of the pre-trained model. Unrelated instances like $X_2$, land far from existing keys and defer to the pretrained layer.

## 2 EDITING LARGE MODELS WITH GRACE

### 2.1 PROBLEM FORMULATION

The Lifelong Model Editing task is to edit the same model hundreds to thousands of times *in a row* without forgetting upstream performance or fixes for previous edits. Assume we are given a large model that was pre-trained on some upstream dataset $\mathcal{U}$. Let $f_0$ denote the frozen pre-trained model at time step $t = 0$. We then deploy $f_0$ on a given downstream task and begin monitoring its predictions $\hat{y}_t = f(X_t)$ as inputs $X_t$ stream in, one step $t$ at a time. Over time, we receive errors $X_t$ one at a time for which $\hat{y}_t \neq y_t$, where $y_t$ is the true label for $X_t$. In order to continue safely deploying $f$, we aim to *edit* $f$ such that $f(X_t) = y_t$. Let $f_t$ denote the *edited* model at step $t$. Note that $f_t$ will be an updated version of model $f$. We desire that $f_t$ maintain high performance on 1) prior edits $X_{<t}$ and 2) the upstream dataset $\mathcal{U}$. Further, upstream training data are often proprietary or prohibitively large, so we assume no access to $\mathcal{U}$ during editing, in contrast to prior works' stronger assumptions (Mitchell et al., 2022a;b; De Cao et al., 2021; Sinitsin et al., 2019; Meng et al., 2022a).

### 2.2 GENERAL RETRIEVAL ADAPTORS FOR CONTINUAL EDITING

GRACE presents a novel paradigm for editing models many times in a row: As errors are identified and corrected over time, GRACE modifies a pre-trained model's behavior *without* altering its weights, as illustrated in Figure 2. A GRACE adaptor at a layer $l$ contains two components: (1) a deferral mechanism that decides whether to use the GRACE adaptor for any given input and (2) a discrete codebook of key–value pairs. The keys are cached activations from layer $l - 1$, denoted by $h^{l-1}$ and the values serve to replace $h^l$, the frozen model's predictions at layer $l$.

**Training and Inference** During training, when a GRACE adaptor at layer $l$ performs an edit, it either creates a new, randomly initialized value, or updates an existing key–value pair. To ensure that values correct the model's behavior, we train the values using backpropagation through the finetuning loss on the model's prediction given the edit. This updated value replaces $h^l$ for the rest of the forward pass. In our experiments, we perform 100 gradient descent steps to train the values.

During inference, if GRACE is activated at layer $l$ – as decided by the deferral mechanism – the value corresponding to the closest key is returned. Similar to the training step, this value replaces $h^l$ for the rest of the forward pass.

**GRACE codebook** A GRACE adaptor at layer $l$ maintains a discrete codebook, adding and updating elements over time to edit a model's predictions. The codebook contains four components:

- *Keys* ($\mathbb{K}$): Set of keys, where each key is a cached activation $h^{l-1}$ predicted by layer $l - 1$.
- *Values* ($\mathbb{V}$): Set of values that are learned while model is deployed and is accumulating errors. Each key maps to a single value – values are randomly initialized and are updated

using finetuning loss on edit examples. By replacing $f$'s activation at layer $l$ with a learned value, $f$ then successfully predicts the correct edit label.

- *Deferral radii* ($\mathcal{E}$): Each key has a *deferral radius* $\epsilon$, which serves as a threshold for similarity matching. Given a GRACE adaptor for layer $l$ and input $h^{l-1}$, we use a similarity search over existing keys to find the key closest to $h^{l-1}$ via a distance function $d(\cdot)$:

$$d_{\min} = \min_i(d(h^{l-1}, K_i^l)).$$

GRACE is activated at layer $l$ *only* if $d_{min} \leq \epsilon_k^l$, where $k$ indexes the most-similar key. The larger the value of $\epsilon$, the more *influence* the key has, since it covers more of the embedding space. As we discuss below, as GRACE edits a model over time, $\epsilon$ values also change, adapting GRACE layers to changing data distributions.

- *Key labels* ($\mathbb{Y}$): When a new key is added, its corresponding edit label is also stored in the codebook. By accessing edit labels *only* while editing, keys and their $\epsilon$ values can be adapted to generalize to similar instances without influencing too much of the embedding space.

**Deferral mechanism** Before editing begins, GRACE layers are empty. As editing progresses, GRACE adds and adapts key-value pairs. Conceptually, performing inference with a GRACE-edited model entails a deferral decision, computing $h^l$ using a discrete key-value search over GRACE's keys:

$$h^l = \begin{cases} \text{GRACE}^l(h^{l-1}) & \text{if } \min_i(d(h^{l-1}, K_i^l) - \epsilon_i^l) < 0, \\ f_0^l(h^{l-1}) & \text{otherwise,} \end{cases}$$

where $f_0^l(h^{l-1})$ denotes the *unedited* model's activation of the $l$-th layer. GRACE $(h^{l-1})$ retrieves the value associated with the key closest to $h^{l-1}$. $\epsilon_i^l$ and $K_i^l$ are the influence radius and key $i$ in layer $l$, respectively, and $d(\cdot)$ is a distance function (we use Euclidean distance in our experiments).

By using a discrete similarity search, if a new input is unlike any cached keys, GRACE simply defers to $f_0$'s pretrained weights. This way, GRACE layers can avoid interference with upstream data by leaving the model unaltered, which can be especially good if input distributions are shifting. Further, if labels flip locally in the embedding space, GRACE partitions the embedding space to correct $f_0$'s predictions *locally*, adapting to the distribution of the input edits. In practice, edits are rare compared to streaming inputs, so a GRACE-edited model will often defer to a pretrained layer's outputs, successfully leaving unrelated inputs unedited.

**Codebook maintenance** As illustrated in Figure 2, when an edit is required, $f_0$ serves as an encoder, computing an embedding for an instance at layer $l$. Then, $f_0^l$ serves as a query across any existing keys in the GRACE codebook for layer $l$. A GRACE layer can perform one of the following operations at any given time step:

1. **KEY-ADD**: If the codebook is empty or the input embedding $h^{l-1}$ falls *outside* the $\epsilon$ radius of all existing keys according to distance function $d(\cdot)$, then initialize a new key $h^{l-1}$ along with a corresponding value $v$, base influence radius $\epsilon^l$, and edit label $y^e$. To alter the model's predictions via the value $v$, finetune the value with respect to the model's final predictions until its prediction is accurate.

2. **KEY-UPDATE**: If $d(h^{l-1}, k_{\text{near}}) \leq \epsilon_i + \epsilon_{\text{init}}$, we must decide whether to expand the influence of the nearest key or to shrink its influence and add a new key, depending on whether or not the keys share edit labels. In practice, we can check if labels are the same by either caching the edit labels associated with each key or simply performing inference. There are two possible casses:

   (a) If $d(q, k_{\text{near}}) \leq \epsilon_i + \epsilon_{\text{init}}$ and the nearest key's edit label is the *same* as the edit label, **expand** the nearest key's $\epsilon$ to encompass the query.

   $$\{K_i^l : [V_i^l, \epsilon_i^l]\} \rightarrow \{K_i^l : [V_i^l, d(f_0^l, K_i^l)]\}$$

   (b) If the nearest key's label is *different* than the edit label, **split** the nearest key by first decreasing the influence radius of the nearest key, then creating a new key-value pair where the key is the query. The new key is simply the query $h^{l-1}$.

   $$\{K_i^l : V_i^l, \epsilon_i^l\} \rightarrow \left\{ \begin{array}{ll} K_i^l : & [V_i^l, \ 0.5 * d(f_0^l, K_i^l)] \\ f_0^l : & [V_{i+1}^l, \ 0.5 * d(f_0^l, K_i^l)] \end{array} \right\}$$

As edits stream in, by continuously adding and updating GRACE's keys and values, the embedding space for a selected layer $l$ becomes partitioned according to which instances need modified outputs. When *not* performing edits, these operations are bypassed, and keys are entirely frozen, regardless of whether or not the instance lands within a key's influence. Overall, GRACE introduces a new model editing paradigm in which edits can be made sequentially, similar edits are encouraged to be edited similarly, and the ultimate influence of new edits can be controlled and monitored explicitly.

### 2.3 GRACE LAYERS FOR SEQUENTIAL INPUTS

When edited layers have sequential inputs, like tokenized sentences, each token receives its own activation. In this case, we broadcast a value to each token's corresponding activations in layer $l$. This approach gives the values strong control over the model's behavior and makes them easier to learn than caching values for the first or last tokens, which may lead to better composability in future works. To find a query's nearest key, we find that euclidean distance works well, similar to (Träuble et al., 2022). GRACE naturally applies to all recent transformer models.

## 3 EXPERIMENTS

To evaluate GRACE, we first show that GRACE outperforms alternative editors by successfully making thousands of edits on a real QA and document classification tasks with shifting labels during deployment. Then, we dig deeper into GRACE, evaluating which layers to edits, effects of hyperparameters, and its generalization when editing a model up to 5,000 times in a row, based *only* on streaming edits. Finally, we illustrate how GRACE makes edits using a simple synthetic example.

### 3.1 EXPERIMENTAL SETUP

We compare GRACE with recent model editors and their capacity to *sequentially edit models hundreds to thousands of times in a row*. This setting is categorically harder than recent works, which make multiple edits *simultaneously*, updating models once based on large sets of edits, akin to finetuning (Mitchell et al., 2022a;b; Meng et al., 2022b). We measure performance for lifelong model editing on multiple axes, including a) performance degradation on upstream data, b) capacity to remember long sequences of previous edits, and c) generalizability to related, but previously-unseen edits.

**Compared methods** We assume no access to training edits, semantically-equivalent inputs, or exogenous datasets at any stage of editing. Therefore, we must modify existing editors for fair comparison. First, we **Continually Finetune** (Lin et al., 2022) on streaming errors. Second, we compare against **MEND** (Mitchell et al., 2022a), which pretrains a hypernetwork on a large set of training edits. We adapt MEND to our setting by continually finetuning MEND's hypernetwork on streaming errors. Third, we implement a **Defer Adaptor**, inspired by SERAC (Mitchell et al., 2022b), which pretrains a deferral model and a prompt-adjusting hypernetwork on training edits, then the deferral model chooses when to trust the pretrained model vs. the editing hypernetwork. We lack training edits, so implement a conceptually-similar adaptor, using a deferral model to predict when to trust the frozen layer vs. predict the next activation with another hypernetwork. Both are continually trained on streaming errors. Fourth, we also consider a soft version of GRACE via a **Memory Network Adaptor**, which includes a memory module and an attention mechanism. The memory module contains learnable values that serve as cached activations, and the attention mechanism takes in the activation from the previous layer (the same as GRACE), then predicts attention weights for each value. The weighted sum serves as the input to the next layer. For all compared editors, we edit only a selected layer. We tune the learning rate for each method, reporting only the best-performing editor for each case. Further implementation details and hyperparameter tuning experiments are available in the Appendix.

### 3.2 GRACE OUTPERFORMS ALTERNATIVE EDITORS ON REAL DEPLOYMENT TASKS

We first compare GRACE to existing model editors on realistic deployment settings by correcting pre-trained models on long sequences of real mistakes on real datasets. To evaluate each model editor, we take a pre-trained model, simulate deployment on a streaming task, and edit the model when mistakes are made. In these experiments, we pass 1,000 inputs into each model and edit only when

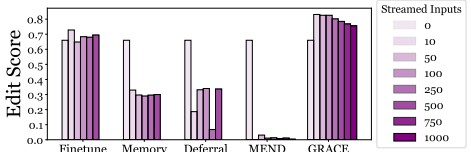

(a) Results from editing a T5 model for open-domain QA with shifting answers from zsRE.

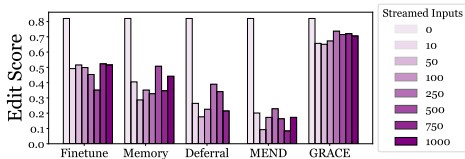

(b) Results from editing BERT for multi-class classification on SCOTUS with shifted labels.

Figure 3: Main results comparing GRACE to alternative editors on context-free QA and SCOTUS document classification. Edit Score is the average of Upstream and Online performance. Each editor is applied to a model deployed on the same sequence of inputs and ends up making roughly 500 edits out of 1000 inputs during deployment. GRACE achieves strong performance both upstream data and previous edits, and $\epsilon$ exerts direct control over this trade-off.

the prediction is wrong. Each editor ends up making roughly sequential 500 edits. Whenever the model is edited, we then measure both Upstream and Online Performance, then compute an Edit Score, which is the average of the two metrics.. In each case, we report results only for methods that succeed to edit the model over 75% of the time, and report once 10 edits have been made; previous editors and finetuning methods are expected to decay upstream performance quickly, since they rely on continually-trained hypernetworks, which should underperform early on without training data. We perform this experiment on two distinct settings using different pre-trained models.

First, we follow Mitchell et al. (2022a) and edit a 60-million parameter T5 model (Roberts et al., 2020) trained for open-domain Question Answering. The T5 model was pretrained on the Natural Questions dataset (NQ) (Kwiatkowski et al., 2019), and we simulate deployment on the zsRE dataset (Levy et al., 2017) extracting potential edits from De Cao et al. (2021)'s validation split. Following Mitchell et al. (2022a), we edit the dense-relu-dense layer of the last encoder block of a 60 million parameter model, though we ablate this choice in Section 3.3.

Second, we experiment with a BERT model on the SCOTUS dataset, which features court documents from across multiple decades. Over the decades, label distributions shift and models trained on earlier data become outdated. Each document is labeled with one of 11 issue areas and SCOTUS is divided into a training split with 7.4k cases from 1946-1982 and a test split with 931 cases from 1991-2016. To illustrate lifelong model editing, we exacerbate the label shifts by merging semantically-similar labels in the training split. For example, we map categories {Civil Rights, First Amendment} → {Civil Rights}. In the test split, we then separate the labels. Further details are in the Appendix.

**Results:** Overall, GRACE dramatically outperforms all four alternative editors, as shown in Figure 3. Over streaming edits, GRACE succeeds to maintain high upstream performance while also remembering how to fix previous edits. We further compare all methods in the Appendix, which includes a hyperparameter study for each method. As expected, continual finetuning is the most competitive with GRACE on both tasks. Also as expected, without access to privileged information, the compared methods decay performance quickly—similar to Hase et al. (2021), we find that after only 10 inputs, the alternatives have already decayed the pretrained model's performance.

### 3.3 MEMORIZATION VS. GENERALIZATION IN GRACE LAYERS FOR THOUSANDS OF EDITS

Next, we evaluate GRACE's memorization vs. generalization performance using extremely long sequences of edits on the QA task. We take the first 2,000 questions from the zsRE dataset, and augment them using 5 rephrasings of each question, resulting in 10,000 questions with which we edit the T5 QA model. Given this controlled setting, we edit each encoder layer's final dense relu module, ranging $\epsilon$ from 0.1 to 10.0, above which performance stabilizes. Each parameter setting leads to the model being edited over 2,500 times, with some being edited nearly 5,000 times[1].

We seek to answer the following questions *simultaneously* in order to explore the tradeoff between memorization and generalization for GRACE by tracking relevant metrics:

---

[1]5,000 sequential edits is extremely large. Hase et al. (2021), for example, perform a sequential editing experiment where they show that even 10 sequential edits catastrophically decays their performance.

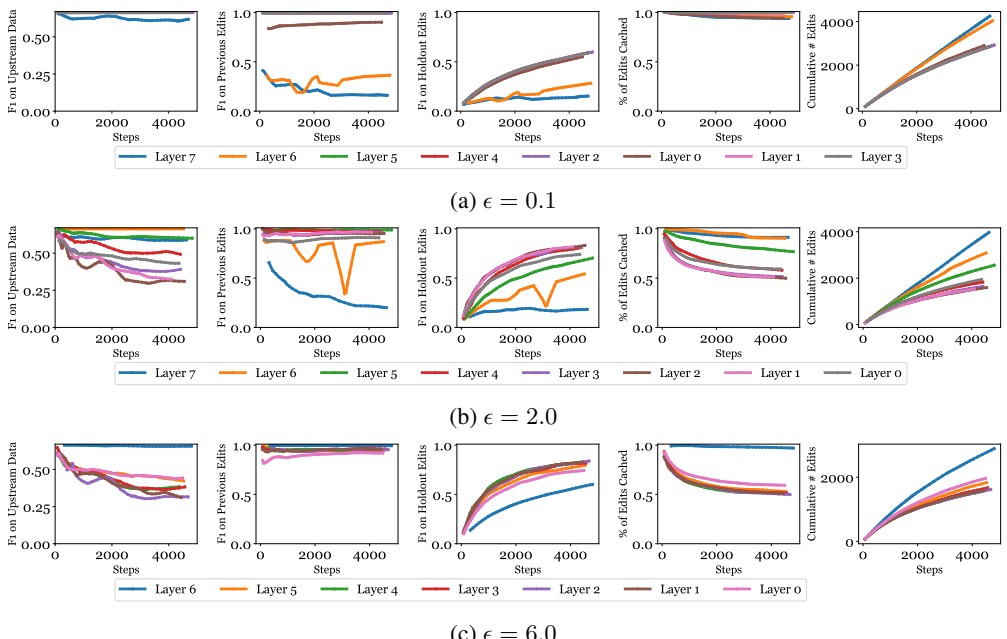

Figure 4: Evaluating GRACE's memorization vs. generalization when choosing $\epsilon$ values and editing different layers. Some layers are better to edit than others, GRACE layers generalizes to unseen edits, $\epsilon$-selection trades off memorization and generalization, and GRACE codebooks stabilize over time.

- How accurate is GRACE for long edit sequences? Metrics: Upstream and Online F1.
- How well do GRACE's keys generalize? Metric: F1 on set of holdout edits.
- Does GRACE simply memorize new inputs? Metric: Fraction of input errors that lead to new keys being created.
- Does GRACE edit every potential edit sample? Tracked metric: Proportion of the 10,000 potential edits that actually require editing

Figure 4 shows our main results, with remaining plots in the Appendix. We derive four key findings:

**1. Not all layers are equally-good to edit**. We first find that GRACE achieves substantially different performance when editing different layers in the T5 model. In Figure 4, we compare the effects of editing the outputs from the final dense ReLU layer of each encoder block. As seen in different colors, the results are strikingly different. First, editing later layers improves upstream performance, but forgets previous errors quickly. On the flip side, editing early layers seems to remember previous edits quite well, but at the expense of upstream performance. For some combinations of layer- and $\epsilon$-choice, there appears to be a good balance. Finally, as shown in the 5[th] column, editing later layers leads to models that need to be edited more often than others, as the number of edits is nearly linear for all three values of $\epsilon$.

**2. GRACE-edited layers indeed generalize to unseen inputs**. We also evaluate generalizability of GRACE-edited models to previously-unseen edits. To achieve this, when a model is edited, we also record its performance on a holdout set of edits containing rephrasings of each training edit. We keep this set of edits static over time, so as edits arrive and are corrected, performance increases on the whole set, as expected. As shown in the 3[rd] column of Figure 4, editing earlier layers leads to better generalization. Excitingly, larger $\epsilon$ values also lead to better generalization, implying that the semantically-similar inputs indeed land in the same deferral radii.

**3. Choosing $\epsilon$ balances memory between upstream data and previous edits**. In Figure 4, we report each metric for different choices of $\epsilon$, the learned deferral radius for a GRACE-edited layer. As expected, we find that for tiny choices of $\epsilon$ (Figure 4a), after correctly choosing the layer to edit, we can achieve nearly-perfect F1 on both Upstream data and Previous Edits. However, this is at the expense of the codebook size: Nearly 100% of edits are cached as new keys (column 4). While such memorization may be sufficient during deployment sometimes, it is akin to a sheer lookup table.

(a) Upstream Training Data   (b) Upstream Training Data + Edits   (c) Predictions before editing   (d) Predictions after editing

Figure 5: Results for GRACE on synthetic data. A model is pretrained on the data in (a), then label shift is introduced in (b) via a local set of label-swapped edit instances. Then, the pretrained model predicts the wrong class for these edit instances, as shown in (c). Finally, as shown in (d), GRACE successfully edits the pretrained model, swapping its labels without impacting distant data.

Excitingly, we find that choosing a bigger $\epsilon$ successfully avoids memorization, while maintaining strong performance.

**4. GRACE codebooks stabilize over time for well-selected parameters**. As shown in the 4$^{th}$ column of Figure 4, we report the % of edits that end up being cached as new keys over time. We find that this % flattens over time, indicating that the rate of key caching decreases as the model trains. As expected, for tiny $\epsilon$ values (Figure 4a), nearly 100% of the edits are cached as keys. However, for larger $\epsilon$ values, the % of cached edits goes down to around 50%, indicating that GRACE adaptors effectively manages codebook size. This finding is true across all $\epsilon$ values and edited layers.

## 3.4   ILLUSTRATING GRACE USING SYNTHETIC DATA

We finally illustrate GRACE using a simple synthetic dataset. As shown in Figure 5(a), we sample two-dimensional instances from two clusters, each corresponding to a class, shown in red and blue. On these data, we pre-train a three-layer fully connected classification network. The network projects the instances into 100 dimensions, 100 dimensions, and back to 1 dimension for binary classification with ReLU activations (Nair & Hinton, 2010). Since these data are linearly separable, the model learns to split the feature space vertically. We then introduce a set of likely edits by introducing a tight circle of instances inside the distribution on the left, simulating local label shift, as shown in Figure 5(b). As shown in Figure 5(c), the upstream model then classifies the flipped labels incorrectly. We then use a GRACE layer to edit the pre-trained model, sampling edits from the edit set one by one and initializing $\epsilon$ to be 0.45 to roughly align with the scale of the edit set.

**Result:** In Figure 5(d), we see that GRACE succeeds to locally edit the model's predictions for the edits without influencing the model's predictions on unrelated training data. While this case is relatively simple, Finetuning a model only on errors—especially those from a single class—leads to catastrophic forgetting on upstream data. We show in the Appendix that Finetuning indeed flips all predictions to the blue edit class. Additionally, GRACE edits this model using *only one key*. Using one key is ideal in this simple setting, since the edits are clustered. As we show next, as tasks become more complicated, edits spread out and require more keys to cover the model's representation space.

## 4   RELATED WORK

**Model editing.** Model editing is a recent and active research area. Classic approaches focus on focus on regularized-finetuning, altering a model's weights by incorporating auxiliary information, like upstream training data and semantically-equivalent instances (Sinitsin et al., 2019). Recent works have extended this paradigm to pretrain hypernetworks that predict edits (De Cao et al., 2021; Mitchell et al., 2022b;a), often decomposing weight updates into low-rank components (Meng et al., 2022a; Hu et al., 2022). Due to notorious training costs, many works focus on editing transformer models (Zhu et al., 2020), leading to natural extensions to multi-lingual editing (Xu et al., 2022). Many recent works have considered parameter-efficient finetuning, often by using low-rank updates Hu et al. (2022). While parameter-efficient approaches like prompt tuning train an extension of pretrained models, they often require more training steps and are more prone to overfit than regular finetuning (Zhong et al., 2022; Su et al., 2022), making them inappropriate from a model editing standpoint compared to approaches such as Mitchell et al. (2022a) and Mitchell et al. (2022b). Some recent

works *have* begun considering editing models multiple times. Batch editing, for instance, attempts to fix models on batches of edits simultaneously. While MEND (Mitchell et al., 2022a) demonstrates their performance decays quickly, more-recent works like SERAC (Mitchell et al., 2022b) and MEMIT (Meng et al., 2022b) are showing more promising results in this direction. However, these exciting works use large amounts of privileged information compared to our problem setting, like computing layer statistics given exogeneous datasets or requiring training sets of representative edits. We assume no access to these extra datasets, and pose that selecting these datasets probably has massive impact on an editor's success. One recent work does discuss sequential editing: Hase et al. (2021) shows that after editing the same model 10 times, editing performance drops dramatically. In our paper, we focus exclusively on this harder sequential editing problem, but edit models *thousands* of times in a row.

Recent works have also begun considering continual finetuning, where large language models are refined over time as new instances arrive. For example, Lin et al. (2022) show that regularizing finetuning with continual learning methods like Elastic Weight Consolidation (Kirkpatrick et al., 2017), Experience Replay (Rolnick et al., 2019), and Maximally Interfered Replay (Aljundi et al., 2019) decay upstream performance rapidly when shifting between multiple tasks.

**Key-Value Methods for Continual Learning.** Key-value methods are now a powerful paradigm for a variety of machine learning problems. These approaches have deep roots in computer vision (Van Den Oord et al., 2017; Liu et al., 2021), driving recent high-profile results like DALLE-2 (Ramesh et al., 2022). Further, key-value methods are particularly strong in continual learning settings, with recent works demonstrating prompt-learning for NLP (Wang et al., 2022b;a) for applications like text retrieval (Xiong et al., 2021). Recent works have shown that *discrete* key-value methods in particular performs well on distribution shifts (Träuble et al., 2022), with recent works extending to question answering (Dai et al., 2022). This performance stems from the stored values, which can remain within the expected distribution of a downstream encoder, regardless of shifting inputs. Further, storing values allows for unlimited long-term memory, resources permitting. We corroborate these advantages in our experiments, where we demonstrate GRACE's robustness to shifting inputs and labels over long sequences of edits.

## 5 CONCLUSIONS

Language models are quickly becoming larger and are being applied to a diverse set of downstream tasks. However, they are often computationally prohibitive to finetune and easily forget past knowledge. In this work, we propose a realistic problem setting where we edit such large models, *Lifelong Model Editing*. In this setting, we edit large models many times in a row given *only* errors that arrive during deployment, without access to any upstream data or test-distribution examples. We then present GRACE, a plug-in module that can wrap around any given layer in any large pretrained NLP model. GRACE layers (1) retain the functionality of the original model, thereby minimizing catastrophic forgetting and (2) adapt to changing data distributions by storing a codebook of cached activations that can grow or shrink over time. We demonstrate GRACE's efficacy by showing that GRACE provides the best trade-off between upstream performance and accuracy on streaming edits among competing model editing baselines and finetuning methods. We further investigate GRACE's capacity to make extremely long sequences of edits and show that choosing the wrong layer to edit can decay performance substantially and that GRACE succeeds to generalize its edits to previously-unseen inputs and avoids simply memorizing edits.

## 6 REPRODUCIBILITY

To encourage reproducibility of both GRACE and our experiments, we have provided thorough descriptions of our experimental setup in Section 3. Our descriptions contain specific layers we choose to edit. In Section 2, we include clear steps that detail GRACE's behavior.

## 7 ETHICAL CONCERNS

Model editing may alleviate some ethical concerns, for example correcting previously-incorrect information. However, editing models may also introduce new errors if the proposed fixes are

themselves incorrect or biased against a given sub-population. Researchers who design, implement, and deploy such techniques should consider how best to evaluate sources of edits and verify they are not harmful.

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

## A    LIMITATIONS

Lifelong model editing is new and challenging, so our method GRACE has limitations. Understandably, introducing codebooks between layers of a pretrained model slows down inference. This slow-down is largely due to repeated similarity searching between embeddings and keys. Even though GRACE's current design does not emphasize inference time, making GRACE faster is a promising avenue for future work, especially given recent accelerations in adaptor methods like LoRA (Hu et al., 2022) and language models using similarity like Gopher (Rae et al., 2021).

While GRACE can scale to use in multiple layers simultaneously, we do not perform these experiments. In the most-general case, GRACE layers may be stacked between subsequent layers. Then, the output of one codebook may be passed to another codebook. Stacking codebooks will lead to highly-flexible models that provide friendly interfaces with which to control their outputs. Our work takes the first step towards this new model primitive, which has clear implications for responsible machine learning.

Finally, autoregressively decoding long sequences with GRACE is particularly challenging. In the long term, language model editors will control long sequence generation. While this is a current challenge for all current editors, GRACE will require modification to generate many tokens in a row. This limitation is due to GRACE's similarity-search between input queries, which are embeddings, and existing keys: During autoregressive generation, the embedding for prior tokens is updated. Then, the resultant embedding may point to new GRACE keys, leading to continuous distribution shift. However, most NLP tasks are not autoregressive, despite autoregressive models currently being popular. We look forward to future work that 1) extends codebook-based edits to autoregressive models, and 2) more-deeply evaluates prior model editors' capacity to alter autoregressive text generation for sentences longer than a few words. In this short text generation paradigm, GRACE can surely be extended.

**Implementation Details**    We experiment with T5 (Raffel et al., 2020) and BERT (Devlin et al., 2018), though GRACE is general and applicable to other transformer-based NLP models as well. To ensure fair comparisons, we edit the same layers across all methods for each experiment and only consider methods that successfully edit each input instance.

For T5 experiments, we edit the dense-relu-dense layer of the last encoder block of a 60 million parameter model, following (Mitchell et al., 2022a). For BERT, we edit a 110 million parameter model's last encoder layer's dense output that was pretrained to classify sentiment. We then finetune BERT ourselves on the SCOTUS training set.

During evaluation, we hyperparameter tune each method, only considering methods that successfully edit each input instance. For Finetuning, Memory Networks, Deferral, and MEND, we edit with learning rates of $1e^{-1}$, $1e^{-2}$, $1e^{-3}$, $1e^{-4}$, and $1e^{-5}$. Intuitively, small learning rates retain upstream performance while large learning rates learn to apply new edits. Therefore, for some learning rates, finetuning methods may achieve higher individual upstream and online scores than we report. However, akin to constrained optimization, the minimal editing success is to actually modify the model's performance on one edit. Here, finetuning is particularly prone to overfitting.

## B    ADDITIONAL DATASET DESCRIPTIONS

**Question Answering with Shifting Answers**    We perform our QA experiments using a T5 model that was pretrained on Natural Questions (NQ). For evaluation, we sample 1000 random question–answer pairs from NQ to serve as upstream data. During editing, for the main results comparing all editors (Figure 3) we extract the first 200 questions that have at least 5 rephrasings in the zsRE dataset. We then use each rephrasing as a separate edit. This creates 1000 potential edits to pass into the pretrained model during deployment. Expanding beyond this setting to evaluate edit generalization presented in Figure 4, we extract the first 1000 instances from zsRE that have at least 10 rephrasings, but split the rephrasings into an edit set and a holdout set. This creates 10,000 potential edits, and 10,000 holdout edits.

**SCOTUS Document Classification**    The US Supreme Court (SCOTUS) is the highest federal court in the United States of America and hears only the most controversial or otherwise complex cases. We consider a single-label multi-class classification task, where given a document (court opinion),

| $\epsilon$ | Num edits | Num keys | key / error ratio | Upstream Accuracy | Accuracy on Prev. Edits |
|---|---|---|---|---|---|
| 1.0 | 403 | 403 | 1.0 | 79.8 % | 99.7% |
| 2.5 | 387 | 256 | 0.66 | 73.9% | 74.2% |
| 5.0 | 375 | 245 | 0.65 | 76.5% | 64.6% |

Table 1: Effect of $\epsilon$ on the number of keys created by `GRACE` for Shifting Annotation Guidelines.

the task is to predict the relevant issue areas. The 14 issue areas cluster 278 issues whose focus is on the subject matter of the controversy (dispute). The list of issue areas include: {Criminal Procedure, Civil Rights, First Amendment, Due Process, Privacy, Attorneys, Unions, Economic Activity, Judicial Power, Federalism, Interstate Relations, Federal Taxation, Miscellaneous, Private Action}. To simulate a scenario for shifting annotation guidelines, we modify the training set as follows:

- Relabel samples in classes {First Amendment, Due Process} as Civil Rights.
- Relabel samples in Unions as Economic Activity.

In the edit set and the test set, we use the original labels i.e. no relabeling was done. Thus, our upstream dataset contains 11 labels and our edit/test sets contain 14 labels, where some classes are expanded in scope as explained above.

## C  ADDITIONAL SCOTUS EVALUATION

We further evaluate the relationship between $\epsilon$ and performance on SCOTUS in Table 1. Here, we see that Upstream Accuracy decays slightly as $\epsilon$ grows, while accuracy on previous edits drops quickly. The `key / error ratio` shows the number of errors that lead to creating a new key. As expected, for $\epsilon = 1.0$, which is relatively small, all edits result in keys. Then, as $\epsilon$ grows, this ratio goes down because inputs land within the deferral radius of existing keys.

## D  MULTI-OBJECTIVE COMPARISONS RESULTS ON QA AND SCOTUS TASKS

We include an alternative view of the results shown in Figure 3 here in Figure 6. In this Figure, we plot each editor's performance on a growing history of previous edits and its performance on the pretrained model's upstream data. For both Figures, optimal performance is in the upper right-hand corner. As expected, each model decays the pretrained model's upstream performance over time, though this decay scales down with learning rate. Also as expected, MEND underperforms in each case as it lacks an edit training set, which is privileged information with respect to our task. We also observe that editing BERT on SCOTUS is far noisier than T5 on QA for all editors. This finding corroborates recent works that show BERT training tends to be highly unstable (Mosbach et al., 2021; Dodge et al., 2020).

We also include the single-metric version of Figure 3 in the main paper. Here in Figure 7, we see that the trends from the main paper remain the same.

## E  ADDITIONAL RESULTS FROM QA HYPERPARAMETER STUDY

To extend the results shown in Figure 4, we add results from more choices of $\epsilon$ in Figure 8. The trends identified in the main paper remain true: some layers are better to edit than others, `GRACE` succeeds to generalize to previously-unseen inputs, $\epsilon$ controls the trade-off between memorization and generalization, and `GRACE` codebooks stabilize in size over time. Performance remains remain the same above $\epsilon = 6.0$. Interestingly, for $\epsilon = 2.0$, Layer 6 appears to be highly unstable over time.

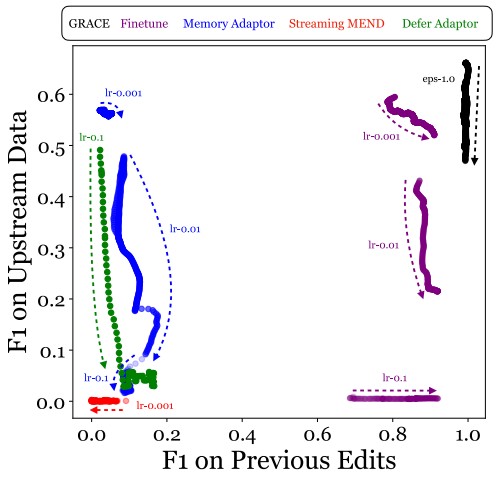

(a) Results from editing a T5 model for open-domain QA with shifting answers from zsRE.

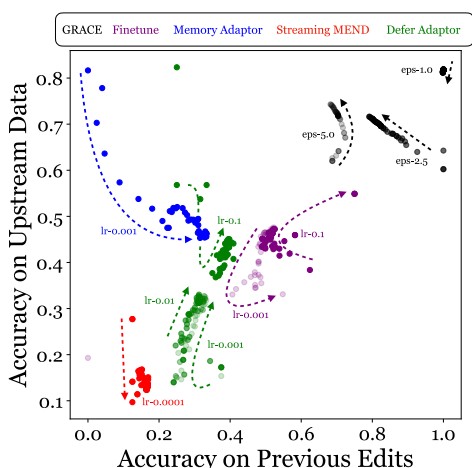

(b) Results from editing BERT for multi-class classification on SCOTUS with shifted labels.

Figure 6: Main results comparing GRACE to alternative editors on context-free QA and SCOTUS document classification. Each editor is applied to a model deployed on the same sequence of inputs and ends up making roughly 500 edits. Ideal performance is in the upper right corner. Arrows denote each model's performance throughout streaming. GRACE achieves strong performance both upstream data and previous edits, and $\epsilon$ exerts direct control over this trade-off.

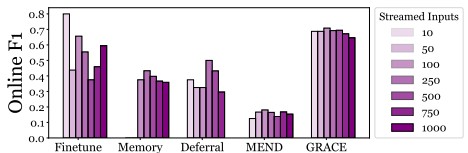

(a) F1 on Previous Edits when editing T5 on zsRE.

(b) Upstream F1 when editing T5 on zsRE.

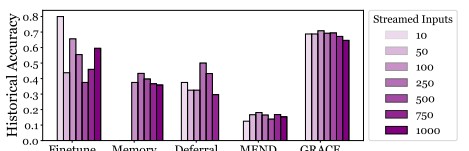

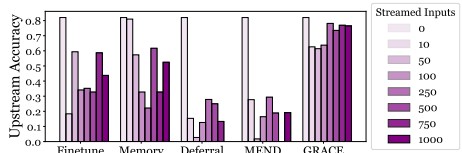

(c) Accuracy on Previous Edits when editing BERT on SCOTUS.

(d) Upstream Accuracy when editing BERT on SCOTUS.

Figure 7: Comparing Upstream and Online performance for GRACE to alternative editors on context-free QA and SCOTUS document classification. Each editor is applied to a model deployed on the same sequence of inputs and ends up making roughly 500 edits out of 1000 inputs during deployment. GRACE achieves strong performance both upstream data and previous edits, and $\epsilon$ exerts direct control over this trade-off.

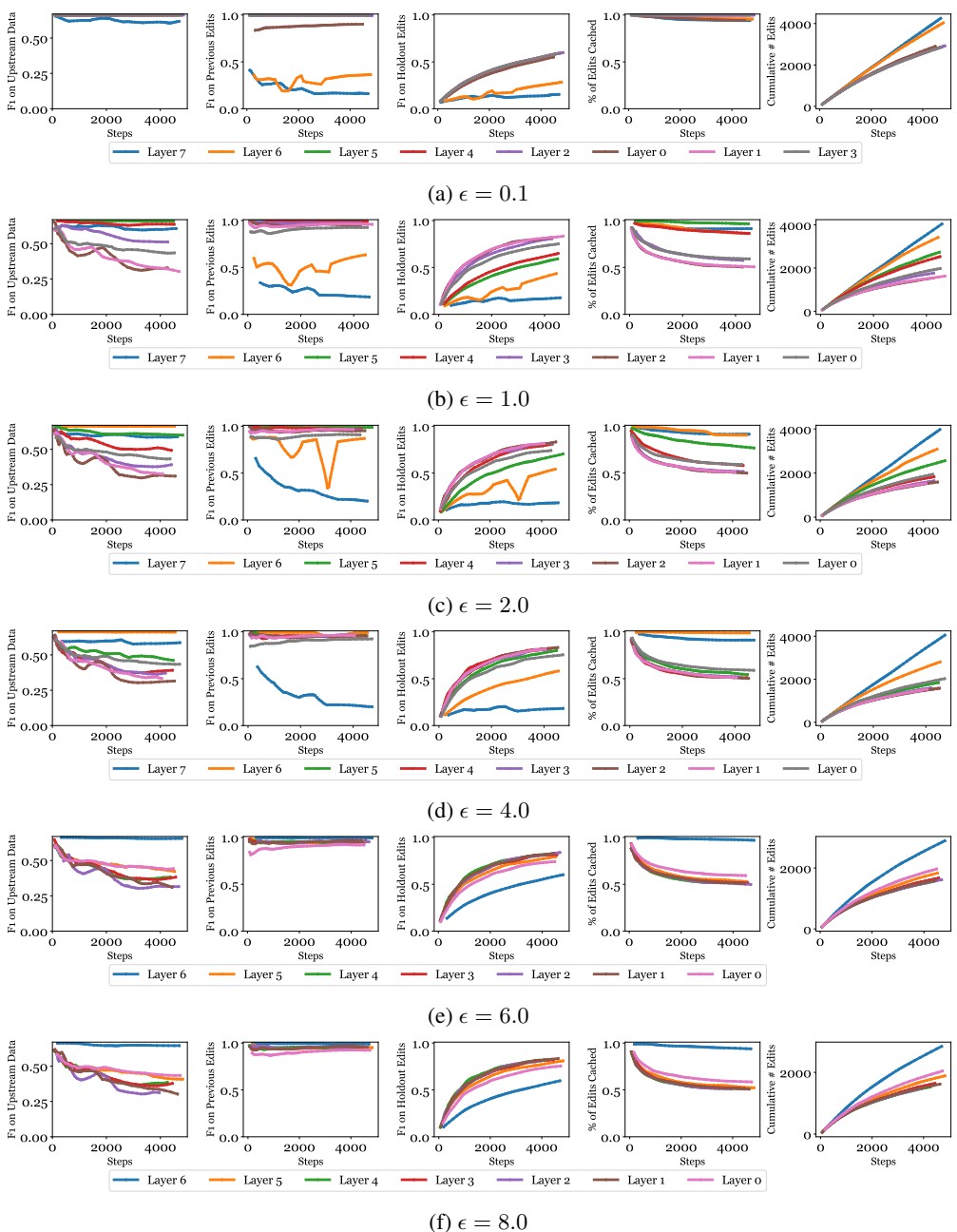

Figure 8: Evaluating GRACE's memorization vs. generalization when choosing $\epsilon$ values and editing different layers. Some layers are better to edit than others, GRACE layers generalizes to unseen edits, $\epsilon$-selection trades off memorization and generalization, and GRACE codebooks stabilize over time.

