# OpenReview forum: "Aging with GRACE: Lifelong Model Editing with Key-Value Adaptors"
_ICLR.cc/2023/Conference — Submitted to ICLR 2023_

### Official Review · Reviewer_WQqG · 2022-10-26

**Confidence:** 4
**Clarity, Quality, Novelty And Reproducibility:** 1. In a nutshell, this paper mainly a…
**Correctness:** 4
**Technical Novelty And Significance:** 3
**Empirical Novelty And Significance:** 2
**Recommendation:** 5

**Strength And Weaknesses:**

Strengths
1. This paper investigates a more realistic streaming setting for model editting.

2. It successfully adapts the discrete key-value bottleneck to the streaming edit setting.

3. Experiments show its superior performances over regular finetuning, streaming MEND, and Memory Net on QA shift and SCOTUS.

Weaknesses
1. The definition of Values is unclear enough. If the key is the output of layer l-1, is the corresponding value used as the input of layer l or l+1? The definition in Section 2.2 says the next layer, which is kind of ambiguous. Figure 2 shows an example of layer l and l+1, which gives the impression of using the output of layer l to retrieve the input of layer l+1. I think it’s necessary to clarify the definitions of keys and values.

2. The part below Figure 2 mainly describes how to add/update keys, with too little information about adding/updating values. Where do the values come from when initializing/adding/updating keys? Are they the same as keys when first added or randomly initialized? How do we update the values to correct the instance error? Do the updates possible affect the previous edits?

3. Section 2.3 doesn’t provide enough training details, what data are used in training and how many gradient steps are used for each edit. Does it use all previous edit data or just the current instance? What if the batch size is too large as the saved edit data increases? This should be an important section to understand how GRACE is trained and used, but the descriptions are limited. I think it’s better to present an algorithm for training and inference.

4. Memory usage? Need to store a large codebook for each layer?

5. Which layers to use GRACE? All layers or some picked layers? If two or more layers use GRACE, does each layer need to do the similar search? If an edit is required, do all layers need to add/update key and values? Slow inference speed?

6. Some typos exist, e.g., “a language model trained in 2016 would correctly Barack Obama as president of the United States,” and “”

7. Ablation studies about which layers to use GRACE and how many optimization steps are used to update the values are missing.


**Summary Of The Paper:**

This work studies how to make large pre-trained models learn to correct errors in deployment. More specifically, it proposes a method to continue editing the pre-trained model in an online streaming setting. The discrete key-value bottleneck is adapted to address the continuous distribution shift. Experiments demonstrate that it outperforms several related methods in the lifelong model editing setting.


**Summary Of The Review:**

This papers adapts an existing technique to a new and interesting model editing setting. Though good performances are achieved, some training details and ablations are missing, which affect the paper quality.

---

> ### Author Response · Authors · 2022-11-19
> **More experiments, method explanations, and training details**
>
> Thank you for your review, we have substantially improved our paper based on your feedback. Please refer to our General Response up above, where we describe the main changes. We specifically implore you to take another look at the methods section (Section 3), which we hope will clear up much of the confusion.
>
> **Values**. In the new version of the methods section, we clarify that the values at layer $l$ serve to replace the activations predicted by the model at layer $l$. The values are trained immediately upon initialization (as described next) and then remain fixed forever to maintain long-term memory.
>
> **Updating Values**. In Section 2, we add details on how values are learned via a finetuning loss on the model. We describe how first, a value is initialized for a codebook in layer $l$. Each value is the same dimension as the frozen pretrained model’s output at layer $l$. After initializing the value, we train it by simply performing gradient descent on its parameters with respect to the model’s finetuning loss according to the new edit label. This way, when the frozen model’s output is replaced by the value, the model will predict the corrected edit label.
>
> **Keys**: We also describe how to initialize the keys, which are simply the inputs from layer $l-1$. Keys then remain fixed forever. However, we do update keys’ deferral radius over time, which determines how close a new query must be to land close enough to the key to trigger the retrieval of its associated value.
>
> **Training Details**. In Section 2.2, we have included further details on training, including that we perform 100 gradient descent steps to train the values right after they are initialized. We agree an algorithm will be extremely helpful. In the Camera-Ready version of the paper, we will include an Inference algorithm, a Training algorithm, and a Key-Value Update algorithm in the Appendix. We are also actively working on a public code release with explicit training configurations.
>
> **Memory Usage**. In our setting, fixing errors is the primary objective. Fortunately, we also show that GRACE only requires a very small number of parameters. In our largest setting, GRACE only consists of ~2% of the pretrained model’s total parameters.
>
> **Which layers to use?** We have extensively studied this question in our new experiments, the main results of which are reported in Figure 4. In this experiment, we find that choosing the right layer is indeed important, and that editing early layers maintains upstream performance while late layers remember previous performance better. Choosing other layers interpolates between these modes, successfully landing on good trade-offs between the key objectives. We leave experiments on editing multiple layers simultaneously to future work. We strongly suspect that multi-layer edits will improve composability of GRACE-edited models.
>
> We have addressed your concerns about details of our proposed method, training details, and experiments about using different layers. We incorporated your feedback into a vastly-improved revision. If you are satisfied with our responses, we ask that you consider raising your score.

---

### Official Review · Reviewer_ngYT · 2022-10-28

**Confidence:** 4
**Correctness:** 3
**Technical Novelty And Significance:** 2
**Empirical Novelty And Significance:** 2
**Recommendation:** 5

**Clarity, Quality, Novelty And Reproducibility:**

- Clarity: The paper is easy to follow.
- Quality: The claims are supported.
- Novelty: The problem is less explored in previous work. The method looks new but is straightforward.
- Reproducibility: It contains informative descriptions, but I am not confident that a knowledgeable reader could replicate the experiment.


**Strength And Weaknesses:**

- Strength: The lifelong model editing setting is an interesting problem setting not explored by previous works. This setting modifies the previous model editing problem to make it closer to a real-world scenario. The proposed approach is simple but effective. This method looks like a reasonable baseline for future works to compare and improve.
- Weaknesses: Readers familiar with model editing may find no surprise in the paper's method and experiments. The setting and method are different from previous works, but readers may have difficulty finding an interesting novelty or new insight that is intriguing or inspiring. I see the contributions made in the paper, but they need more depth, like the previously accepted papers have. One way to improve is to beef up the experiment section. Ex: create easy/hard streaming editing scenario, ablation study, level of distribution shift versus editing success rate, etc. Lastly, the SOTA method [1] should be included and compared. [1] looks to have very good potential in the lifelong model editing setting.
- Other concerns/questions:
1. In Figure 4(a), why does stream MEND start from such a low upstream F1? Should every method start from a point close to the unedited model f0?
2. Although MEND was not designed for streaming model editing, its success rate is expected to be reasonably good for the first few editing in the stream. However, Figures 4(a) and 5(a) show a very poor result of MEND/fine-tuning from the beginning. Why?
3. In Figure 4(a), the colors for different hyperparameters are not shown in the figure. Only the darkest color is visible (ex: light purple and light red do not appear in the figure). This could need to be clarified for readers.
4. limitation in applying autoregressive decoding. I see the author mentioned this in its limitation section, and I appreciate it. However, this is a huge limitation that does not occur with other model editing strategies.
5. About the Memory Network baseline, is there a reference paper? Please provide more detailed descriptions if it is created for this paper.

[1] https://arxiv.org/abs/2206.06520

**Summary Of The Paper:**

This work addresses the lifelong model editing setting, where errors stream into a deployed model, and the model is updated to correct wrong predictions. The author uses a key-value strategy to look up the codebook of edits to change the behavior of a model. The input feature of a module is used as the key, while the desired output of the module will be the value stored in the codebook. The edits are stored as key-value pairs and extracted based on the similarity between an input feature and the stored keys. The experiment uses QA datasets (zsRE) and document classification data (SCOTUS) to create the stream editing scenario and compares it with MEND, fine-tuning, and a memory-based method.

**Summary Of The Review:**

The problem setting is interesting, and the method is marginally novel. The results look good but need more analysis to provide in-depth insight into the problem. Overall the weakness outweighs its strength.

[Post-rebuttal]

The evaluation has considered all revisions and responses. This form was up-to-date.

---

> ### Author Response · Authors · 2022-11-19
> **More experiments, method explanations, and training details**
>
> Thank you for such detailed feedback, please see our General Response up above, which highlights **new, beefed up experiments with more comparisons on more settings**.
>
> **Additional comparisons**. SERAC is not directly comparable in our setting due to its requirements for privileged information regarding our problem. However, we have added a new baseline inspired by SERAC’s key functionality: A Deferral Adaptor that chooses when to trust the frozen, pretrained model. We have added a more-detailed description of this method in Section 3.1, where we describe how the Deferral Adaptor uses a continually-trained hypernetwork that predicts whether to trust the frozen layer vs. predict the next activation with another hypernetwork. Below, we show the average Edit Score (mean of upstream/online performance) for each method (summarizing Figure 3). The Deferral Adaptor outperforms MEND (like SERAC does), but is outperformed by GRACE.
>
>  | | MEND | Deferral | GRACE |
>  | ------------------- | -------------- | ----- | -
>  | zsRE | 0.018 | 0.256 | 0.75 |
>  | SCOTUS |  0.162 | 0.23 | 0.65 |
>
> **Pre-edit performance before editing on QA**. Our results show performance after already editing 10 times. This decayed performance is expected, since MEND has no access to its needed privileged information and therefore overfits the model updates to the early errors. This rapid decay is corroborated in the MEND paper and in Hase et al., 2021, where editing decays models very quickly, even when they have privileged information.
>
> **Figure Clarity**. Thank you for pointing this out, we agree and have replaced Figure 3 with a clearer depiction of our results, highlighting the best-performing hyperparameter settings. We moved the more-detailed version of the figure to the Appendix, where we clearly annotate the hyper-parameters.
>
> **Autoregressive Decoding**. We generally agree that autoregressive decoding is an important modeling paradigm. However, not all models are autoregressive and for classification problems, which are arguably more practical, our method is a clear match. We look forward to extending GRACE to work autoregressively, but this is out of scope for the contributions of this work.
>
> **Further baseline details**. We have revamped our experiments section and included clearer descriptions of our baselines in Section 3.1, in which we clarify that the Memory Adaptor includes a memory module and an attention mechanism. The memory module contains learnable values that serve as cached activations, and the attention mechanism takes in the activation from the previous layer (the same as GRACE), then predicts attention weights for each value. The weighted sum serves as the input to the next layer.
>
> We have addressed your concerns about experimental depth and compared methods. We have also uploaded a substantially-improved revision based on your feedback. If you are satisfied with our responses, we ask that you consider raising your score.

---

### Official Review · Reviewer_PaQG · 2022-11-01

**Confidence:** 4
**Correctness:** 3
**Technical Novelty And Significance:** 2
**Empirical Novelty And Significance:** 3
**Recommendation:** 6

**Clarity, Quality, Novelty And Reproducibility:**

The writing was mostly clear, though some details were missing. In particular, the details of the datasets used during evaluations (how were edits sampled during editing and while evaluating whether previous edits are preserved, what do they look like, etc.) as well as some detail about baselines (like the memory network) made interpreting the experimental results a bit difficult. Also, the procedure for actually producing the "values" in Figure 2 isn't explained until the top of page 5, which made me think I'd missed something for a while (this is a relatively minor point, though).

Other points of confusion:

- *“However, large pools of training edits are rarely available before deployment, otherwise the edits could be used during pretraining.”* This statement doesn’t really make sense; trainable model editors train on a finite set of training edits to learn the ***general behavior of editing***, not to learn the specific information in those edits. Existing evaluations of model editors (e.g., Mitchell et al) evaluate the editor on edits that do not appear in the editor training set. So including that information in the pre-training set wouldn’t help; the specific training procedure of learning a model editor is necessary.
- With the given criteria for a successful edit, it seems like the optimal epsilon is just arbitrarily small. Table 1 supports this claim, and contradicts the statement in the paper: “As we increase ε, we begin to see that GRACE trades off upstream accuracy for online accuracy” From my reading, as we increase eps, both metrics get strictly worse; there’s no tradeoff.
- *“Additionally, prior model editors have yet to consider sequential”* This isn’t true; see [1] and possibly Mitchell et al., 2022b (in which batched edits and sequential are somewhat equivalent)

In terms of novelty, the editing technique is somewhat novel, though fairly closely related to the LU baseline in Mitchell et al., 2022b and the ROME method in Meng et al., 2022. The problem setting is not novel, as sequential editing is considered in [1].

[1] Hase et al., 2021. Do Language Models Have Beliefs? Methods for Detecting, Updating, and Visualizing Model Beliefs.

**Strength And Weaknesses:**

Strengths:
- Focuses on an interesting extension of existing model editing work, which has mostly focused on single/batched model edits, rather than streams of edits
- The proposed method does not need a training set of edits, unlike several past model editors. This requirement is a non-trivial limitation of some past works.
- The proposed method is able to apply many edits without interference or forgetting of the pre-training data

Weaknesses:
- Baselines are limited and not explained in detail (particularly the "memory network" baseline). The setup for MEND is a bit strange, and it’s not too surprising that it fails. Why not compare with ROME (Meng et al., 2022), which I believe doesn’t require an editing training set?
- The criteria for successful edits are, in my opinion, insufficient, in that they do not consider the generalization of the edit. Previous work such as de Cao et al., 2021, Meng et al., 2022 and Mitchell et al., 2022a,b evaluate the generalization of an edit to related inputs in some manner, which is crucial in my opinion.
- On a related note, it’s not clear to me the extent to which the evaluations test the generalization of an edit; Section 4.2 says “For editing, we sample 1000 edits from zsRE, including random samples of 5 rephrasings for each question.” Does this mean the the F1 on previous edits is computed using the rephrasings of the original edit X? If we’re not evaluating generalization in some way, it’s not clear that we need to cache representations at all, vs just caching the raw inputs and labels? My understanding is that the advantage of existing learnable model editors like e.g. Mitchell 2022b is that the model can learn to reason over edits to produce significantly different outputs.
- The experiments are lacking in qualitative examples; it would be helpful to analyze some success and failure cases to see where the proposed method begins to fail (e.g., with respect to generalization).
- The method is not particularly technically novel (minor point)

I believe the paper would be improved by including additional model editing baselines, such as ROME (Meng et al., 2022), SLAG (Hase et al., 2022), or MEND using a proper edit training dataset (this would give MEND privileged data, but would be helpful to see if MEND is even able to apply sequential edits in the best possible scenario; maybe it still fails). In addition, it's critical that experimental evaluations clearly evaluate the generalization capabilities of the model editor, one way or another. It's not clear how the currently-reported metrics do so. Otherwise, a "successful" editor could be one that purely performs memorization (i.e. a simple dictionary/lookup table mapping from input to label).

**Summary Of The Paper:**

This paper proposed a new method for *model editing* in pre-trained language models, which has received growing attention from the deep learning community in the past few years. The authors point out that the setting of sequential model editing (in which the model must be edited with a stream of edits, rather than a small/static set of edits) is relatively understudied, and propose GRACE, a method that is better-suited to applying many model edits without destroying model performance on the pre-training data or forgetting past edits. Under a their particular evaluation conditions, the authors find their method is better able than some existing methods to handle many sequential edits.

**Summary Of The Review:**

Overall, I think this work is getting at an important question for folks interested in model editors: how do we apply many edits to our model without destroying it. However, I feel the work as-is is lacking in experimental rigor (in terms of both baselines considered and the specific quantities measured in the experiments; i.e., generalization) as well as technical insight and novelty. Therefore, I would recommend rejecting the paper in its current form.

**Update** in light of author response:

I'm grateful for the authors' thorough response to my concerns; I think the paper is significantly improved. The additional generalization ablation is helpful in understanding how well GRACE generalizes to difference expressions of an edit. My reading of the result is that GRACE does some generalization to paraphrases of the edits, but still fails to generalize to a significant number of simple paraphrases of the edit content (as shown by the F1 score in column 3 of Figure 4 being significantly below 1). Increasing the deferral radius helps with this issue, but comes at the cost of significantly reduced upstream F1.

My remaining concerns about GRACE as a method are related to this generalization issue. GRACE seems to generalize only mostly to paraphrases of edits, and I would guess it is totally incapable of generalizing to entailed edits, such as:

Edit:

Where is the Eiffel Tower located? Rome

Test input:

What famous tower is located in Rome? <should answer Eiffel Tower>

It seems like editors like SERAC (Mitchell 2022) would be able to perform this type of generalization, while GRACE's explicit key-value construction makes it extremely unlikely that such generalization would occur, given that it already struggles somewhat with paraphrases.

As a related generalization issue, GRACE's reliance on exact equality of the edit label to determine if edits contain equivalent information is problematic. Consider the edits:

e1: What is the tallest mountain on earth? Everest
e2: What is the earth's tallest mountain? Mount Everest

Reducing the radius of e1 to make room for a separate edit e2 feels wrong to me. How do the authors propose handling the problem of determining the semantic equivalents of edit labels (or edits, in general)?

In conclusion, I'm now actually okay with accepting the paper in that it is essentially a novel combination of memory-based (e.g. SERAC, Mitchell 2022) and parametric (e.g. ROME, Meng 2022 or MEND, Mitchell 2022) editing ideas that shows some promise. However, I am also okay with rejecting the paper this time around, on the grounds that GRACE may not be fully leveraging the capabilities of existing methods, producing a sub-optimal generalization-upstream performance frontier, and the positive result of applying many edits is largely offset by the weaker ability of GRACE to generalize the content of the edit to related inputs (paraphrases or entailed information).

---

> ### Author Response · Authors · 2022-11-19
> **New generalization experiments, methods descriptions, and clarifications**
>
> Thank you for such detailed feedback, we have substantially improved our paper based on your feedback. Please refer to our General Response up above, where we highlight **new experiments with more comparisons**, showing that **GRACE generalizes to holdout edits**. Below, we address your specific concerns.
>
> **Edit Generalization and Memorization**. We agree generalization experiments are needed, so we have conducted and added a new experiment in Section 3.3 of the revised paper. We have **summarized the new findings in the General Response** up above, and we implore you to take a look at the revised paper for detailed descriptions. We find that GRACE indeed generalizes to previously-unseen edits and that GRACE makes edits without simply memorizing previous edits, given good selections of $\epsilon$ and chosen layers to edit.  We have also added clearer dataset descriptions to the Appendix, where we clarify that in the main result, we evaluate on exact previous edits (not rephrasings), but in the ablation we add results for holdout, previously-unseen edits, which are rephrasings of edits.
>
> **Memory Adaptor Description**. We have added more straightforward descriptions of each baseline in Section 3.1, where we clarify that the Memory Network Adaptor maintains a fixed-size set of values and an attention mechanism that accesses them as a function of the input query (the same query as in GRACE). Given the query, the attention mechanism applies weights to each value, then takes a weighted sum over all values, and the weighted sum serves as the activation to be passed into the next layer. The values are learned over time, and different activations are retrieved according to the input query.
>
> **Extra Comparisons**. ROME is not directly comparable since it 1) applies only to GPT without clear extension to alternative models, and 2) requires an exogenous dataset to compute layer statistics during editing.
>
> **Clarifications**
> * “Edits could be used during pretraining”: Thank you for pointing this out, we agree that simply finetuning the pretrained model on edits may not always lead to better models, so we removed this claim. However, in our paper’s context, our point still rings true: access to training edits is the same as access to the distribution of the model’s potential future mistakes. We believe a major benefit of model editing is to fix errors during deployment,  and access to well-predicted future mistakes is highly-unrealistic in many real deployment tasks.
> * “Optimal epsilon appears to be arbitrarily small”: Our new experiments clear this up, and it appears that on the SCOTUS dataset, small epsilons are indeed strictly better according to Upstream and Online Accuracy. However, this is at the expense of codebook size (see Figure 4), which may or may not be constrained.
> * “Prior model editors have yet to consider sequential”: Thank you for pointing us to Hase et al., 2021 which indeed discuss sequential editing and includes a sequential experiment. We have added their work in our related works section. However, Hase et al., 2021’s is still far from our problem setting: they are performing Knowledge Editing, requiring privileged information which assume no access to, then appear to only perform 10 sequential edits (our experiments show up to 5000 sequential edits).
> * *Batched vs. Streaming editing (our setting)*. Batched editing is not the same as sequential editing. In batched editing, the model applies many edits simultaneously, possibly sharing  information across batch elements, then the updated model is evaluated. In contrast, we use streaming errors, one at a time, and aim for a successful model after every edit. Both are realistic and important, but they are different.
> * We agree qualitative examples will help with clarity, so we will add some to the camera-ready paper’s appendix.
> * Thank you for pointing us to LU, we agree it is similar. However, by learning our values, GRACE will lead to more-general edits, and GRACE adapts to changing distributions of edits by updating $\epsilon$ over time.
>
> We have addressed your concerns about generalization, compared methods, novelty, and dataset descriptions. Based on your feedback, we have substantially improved our submission.If you are satisfied with our responses, we ask that you consider raising your score.

---

### Author Response · Authors · 2022-11-18
**General Response**

Thank you to all reviewers for providing such extensive, high-quality reviews. We are happy to see that all reviewers agree that our problem is interesting and important, and that our realistic setting is not deeply explored by previous works. Given how actionable nearly all comments were, we have uploaded a substantially improved version of our paper. We address each of your concerns individually below, but first we highlight the major revisions to the paper.

**Restating Problem Definition for Clarity**. We edit models hundreds to thousands of times in a row based only on singular errors immediately when they arrive. Every time the model is updated, its performance should remain high. We assume no access to any upstream data and we also assume no access to any training edits, which is infeasible in many applications of our setting: Edits are often mistakes, so we will rarely have a set of known likely-to-be mistakes before deployment. Having such information would be akin to knowing how your distribution will shift beforehand. We also assume edits to be simple: Given X, my model should have predicted y. This is broader than editing factual knowledge. Our setting is also not batched editing, where many edits are made simultaneously. Both settings are realistic in different scenarios.

**Extensive New Evaluation**. We have included an entire new experiment in Section 3.3.
We perform this experiment for extremely long sequences of edits, with some models getting updated up to 5,000 times in a row. We list the main objectives of the experiment along with our key-findings:

**Does GRACE’s generalize to previously-unseen edits?**
* Yes: GRACE achieves an F1 of nearly 0.85 on a new set of completely-unseen holdout edits

**Which layers should you edit?**
* Some layers are better to edit than others. For instance, in zsRE QA (Section 3.3), editing later layers achieves better upstream performance, while editing earlier layers is better for online performance for most choices of $\epsilon$.

**What is the effect of tuning epsilon?**
* Tuning epsilon indeed trades off upstream performance, edit history, generalizability, and codebook size

**How does the size of GRACE codebooks change over time?**
* GRACE codebooks stabilize in size over time

We also note that previous methods such as MEND/ROME/SERAC have generally targeted applications that require updating factual knowledge that can be verbalized through text (e.g., QA, factoid checking). It is not at all clear whether/how such methods could generalize to broader applications that we consider in our problem setting (e.g., our shifted label experiments). We have tried to adapt these methods to our setup whenever possible (e.g., MEND is one of our baselines), but we note that some of these baselines are not straightforwardly applicable (e.g., it is unclear how ROME can be used for classification).

---

### Decision · Program_Chairs · 2023-01-20

**Decision:**

Reject

**Justification For Why Not Higher Score:**

The new experiments that test generalization do not include any comparison to prior methods (i.e. only ablations), and the lack of thorough generalization experiments is a key weakness

**Justification For Why Not Lower Score:**

n/a

**Metareview: Summary, Strengths And Weaknesses:**

This paper introduces some interesting new ideas that address a relevant problem, but there is not enough reviewer support to accept this paper. The author response significantly improved the paper, but it does not quite go far enough. In particular, the new experiments that test generalization do not include any comparison to prior methods (i.e. only ablations), and the lack of thorough generalization experiments is a key weakness. I'd encourage the authors to use all of the feedback to improve the paper and resubmit to a future venue, since the paper does have promise.